# Opportunities and Challenges of Data-Driven Virus Discovery

**DOI:** 10.3390/biom12081073

**Published:** 2022-08-04

**Authors:** Chris Lauber, Stefan Seitz

**Affiliations:** 1Institute for Experimental Virology, TWINCORE Centre for Experimental and Clinical Infection Research, a Joint Venture between the Hannover Medical School (MHH) and the Helmholtz Centre for Infection Research (HZI), 30625 Hannover, Germany; 2Division of Virus-Associated Carcinogenesis (F170), German Cancer Research Center (DKFZ), 69120 Heidelberg, Germany; 3Department of Infectious Diseases, Molecular Virology, University of Heidelberg, 69120 Heidelberg, Germany

**Keywords:** virus discovery, computational virology, virosphere in health and disease, sequencing archives, data mining

## Abstract

Virus discovery has been fueled by new technologies ever since the first viruses were discovered at the end of the 19th century. Starting with mechanical devices that provided evidence for virus presence in sick hosts, virus discovery gradually transitioned into a sequence-based scientific discipline, which, nowadays, can characterize virus identity and explore viral diversity at an unprecedented resolution and depth. Sequencing technologies are now being used routinely and at ever-increasing scales, producing an avalanche of novel viral sequences found in a multitude of organisms and environments. In this perspective article, we argue that virus discovery has started to undergo another transformation prompted by the emergence of new approaches that are sequence data-centered and primarily computational, setting them apart from previous technology-driven innovations. The data-driven virus discovery approach is largely uncoupled from the collection and processing of biological samples, and exploits the availability of massive amounts of publicly and freely accessible data from sequencing archives. We discuss open challenges to be solved in order to unlock the full potential of data-driven virus discovery, and we highlight the benefits it can bring to classical (mostly molecular) virology and molecular biology in general.

## 1. From Technology-Focused to Data-Driven Virus Discovery

The way viruses are identified has changed fundamentally since the first viruses were discovered at the end of the 19th century due to technological advancements. The first discovery of a virus, now known as tobacco mosaic virus, by Dimitri Ivanovsky in 1892 [1] and its independent validation by Martinus Beijerinck in 1898 [2] was based on using porcelain Chamberland filters invented in 1884, which retain bacteria and other microorganisms, but not viruses [3]. The discovery of the first animal virus, foot-and-mouth disease virus, by Friedrich Loeffler and Paul Frosch followed shortly after [4]. The technologies available at that time provided evidence for virus presence, but not for virus identity. It was electron microscopy and X-ray crystallography that enabled researchers to prove that viruses form particles and to resolve the structure of these particles in the first half of the 20th century [5]. Likewise, the invention of Sanger sequencing by Frederick Sanger and colleagues in 1977 [6] and the polymerase chain reaction (PCR) by Kary Mullis in the 1980s [7] paved the way to identify viruses via their nucleic acid sequences, bringing virus discovery to a new level in terms of speed and scalability. It was now possible to accurately determine virus identity. The invention of high-throughput approaches for massively parallel sequencing and long-read sequencing, also known as second- and third-generation sequencing, has further transformed virus discovery, particularly when combined with metagenomics and metatranscriptomics approaches, and is producing an ever-increasing amount of sequence data. Nowadays, the genome sequence, either partially or completely determined, is the only biological information available for the vast majority of viruses. Consequently, bioinformatics data analysis, although already assisting virus discovery and virological research in general in the pre-sequencing era, is playing a pivotal role in sequencing-based virus discovery.

Irrespective of the technology used, researchers aiming to discover novel viruses were dependent on the availability or collection of biological samples and their analysis in wet labs, the latter involving DNA/RNA extraction, library preparation, and sequencing [8,9,10,11,12]. This conventional virus discovery approach has been highly successful and is particularly promising for discovering undescribed viruses in poorly characterized and understudied hosts, as well as in environmental samples. Notably, however, the requirement of access to biological samples has been lifted recently. It has been demonstrated that data from publicly and freely accessibly repositories of assembled, but largely unannotated, sequence data from the Transcriptome Shotgun Assembly (TSA), the Whole Genome Shotgun (WGS), and the Integrated Microbial Genomes & Microbes (IMG/M) databases [13,14], as well as raw sequencing reads, first and foremost from the Sequence Read Archive (SRA) of the U.S. National Center for Biotechnology Information (NCBI) [15], present a unique source of both known and novel viral sequences that can be exploited efficiently [16,17,18,19,20,21,22,23]. Importantly, these approaches combine virus identification with the quantification of inter-virus relations via comparative genomics and phylogenetics to define virus novelty within the taxonomically structured sequence space of the virosphere. They rely on advancements in computing, concerning both hardware and software. We propose that this new approach to virus discovery, which is sequence data-centered and primarily computational, and to which we refer as Data-Driven Virus Discovery (DDVD), has unique potential for exploring the natural diversity of viruses that exist on our planet in unprecedented detail and depth. We expect DDVD to bring various benefits for whole virology and beyond, and below, we discuss associated opportunities and challenges, focusing on SRA-based DDVD.

## 2. Opportunities Brought by Data-Driven Virus Discovery

### 2.1. The SRA as a Unique Source of Viral Sequences

An advantage of DDVD over conventional virus discovery approaches that cannot be overestimated in our view is the amount of available data, which outcompetes by a large margin that of any conventional virus discovery data set. As of June 2022, NCBI’s SRA (and its mirrors at the European Bioinformatics Institute (EBI) and the DNA Data Bank of Japan (DDBJ)) included data for more than 10.4 million sequencing experiments conducted on 8.6 million different biological samples. About 8.3 million of the experiments were on eukaryotes, and the large majority of the data sets (90.7%) were publicly accessible, while the remaining <10% were under controlled access. Although about 3.2 million experiments were of human origin, the SRA contained data for about 119,000 different species, with 70.8% of them being eukaryotes. The accumulation of new sequencing experiments in the SRA during recent years has proceeded at a rate that strongly exceeds linear growth, with a size increase of 21.4% just within the last year and a current doubling period for the amount of open access data of approximately two years (Figure 1). This invaluable resource comprises data from a significant fraction of species that exist on Earth, and the members of each of those may be hosts of known or unknown viruses. Indeed, it has been shown that gene or genome sequences from both endogenous and exogenous viruses can be detected in nucleotide archives as a by-product of sequencing the host, indicating that many of the organisms for which data have been deposited in the SRA were infected by one or several viruses by the time of sampling [18,23,24]. Typically, these studies with “viral stowaways” were unrelated to virus research and the presence of viral sequences went unnoticed by the original authors that produced the sequencing data. Retrospective detection of viral sequences may be possible for the majority of data sets in the SRA [23], and continuous mining of sequencing data from the SRA and similar databases is, therefore, expected to gradually expand our knowledge of the natural genetic diversity of viruses.

### 2.2. High-Throughput Mining of Raw Sequencing Data

Notably, it has been demonstrated that the vast amount of unprocessed data in sequencing archives can be analyzed efficiently through highly parallelized computation using high-performance computing [20,24] or cloud computing [23] platforms. This allows for the fast (re-)evaluation of old and newly deposited data sets at regular intervals. It will be interesting to see whether the accumulation of newly discovered viral genomes will saturate in the near future. In this respect, it will be important to understand whether the increasing number of known viral genomes will enable the discovery of highly divergent viruses that have so far escaped sequence similarity-based detection (but see also below for associated challenges). If DDVD can close major gaps in our knowledge of viral diversity, as has been suggested recently by the discovery of members from several potentially novel RNA virus phyla [22,25], it offers a promising avenue toward a comprehensive description of the virosphere.

### 2.3. Benefits for Biological and Medical Sciences

Conventional virus discovery studies typically look for viruses (i) in sick hosts, focusing on predefined pathogenic viruses or virus groups, (ii) in economically important hosts, or (iii) in often poorly characterized hosts living in a geographically well-defined area, for instance, at the human–wildlife interface of a certain country or state. The discovery of viruses via DDVD in samples collected for other purposes can offer a fresh perspective for virus diversity research and beyond. For instance, the uncontrolled (and usually unnoticed) presence of viruses may account for unexplained variance in the outcome of experiments of the same kind involving a certain host by different laboratories or the same laboratory over a longer period of time. DDVD enables the identification of such missing variables and, therefore, offers a way to connect to researchers of various expertise outside virology. Taking this reasoning another step forward, DDVD could stimulate the field to develop standards that enable the identification and (genetic) characterization of all organisms in a biological sample, including bacteria and other microorganisms, even if only incomplete sequences can be obtained, which could, nevertheless, provide valuable information [26,27].

### 2.4. Host Assignment

The availability of often very detailed metadata presents another notable strength of DDVD. In the case of the SRA, information about the organism (at the species or genus taxonomic rank) from which the sequencing sample was obtained, as well as tissue or even cell type annotation, is typically available. This offers the opportunity to assign the host and possibly organ tropism of the newly discovered viruses. Moreover, we expect that the confidence of host assignment, at a reasonable and useful taxonomic rank, can be further increased by taking into account viral phylogeny, as closely related viruses generally tend to have similar hosts, particularly in the case of viruses infecting eukaryotes [18]. Host assignment is more complicated for uncultivated bacteriophages, but bioinformatics methods tackling this challenge are being developed as well [28]. The task of confidently predicting viral hosts requires the ability to detect cases where the viral sequence originated from any kind of contamination, which may have happened during sample collection, sample preparation, or the actual sequencing, and could originate from various sources, including contaminated laboratory components or index hopping [29,30,31]. We acknowledge that the confident detection of contaminants in sequencing data is currently a very difficult task. We hypothesize, however, that the tremendous number of metadata-annotated viral sequences that will be available for analyzing possible correlations of viral and host phylogenies will facilitate the discrimination between origins by infection and contamination within a statistical framework. Guidelines for confident host assignment in DDVD studies may include (i) the detection of a viral genome in sequencing experiments from at least two independent laboratories, (ii) the determination of a significant part of the viral genome necessary for accurate phylogenetic placement, and (iii) sufficiently deep read coverage of the viral genome to discriminate between actively replicating viruses from contaminants in at least one instance.

### 2.5. Data Access and Accuracy

Additional opportunities of DDVD that should not be overlooked relate to the data it relies on. Firstly, we emphasize that the usage of these data for scientific purposes is free of financial costs beyond standard investments in IT infrastructure that any laboratory has to make. The whole scientific community has already spent billions of USD (conservatively assuming an average cost of at least 100 USD per sequencing experiment) to generate the data in the SRA and similar repositories, which can now be re-used freely and accessed publicly. The analysis of the data will also not induce additional costs if scientific high-performance computers are utilized, or assuming that charges normally made by cloud-computing providers can be circumvented [23]. Second, although also applicable to conventional virus discovery approaches and thus not exclusive to DDVD, is the fact that sequencing data are highly accurate, with an error rate of around 0.1% per nucleotide on average in the case of Illumina-based platforms [32], discriminating them from many other types of data in molecular biology. The accuracy of the more noisy long-read sequencing technologies is improving as well [33]. Additionally, the SRA metadata are commonly of high quality. Therefore, DDVD builds upon an unparalleled foundation of a vast amount of highly accurate, often well-annotated data that can be accessed at no charge by virtually anyone from anywhere in the world.

## 3. Challenges to Be Solved That Facilitate Data-Driven Virus Discovery

### 3.1. Assembly Quality Standards

In order to realize the full potential of DDVD and maximize its impact on virology, several main issues require the development of a community-wide consensus. We believe that it will be prudent to generate and apply quality standards for viral genome assemblies produced by DDVD studies to ensure their reliability and acceptance by the virology community. We consider this to be of high relevance because of the exceptionally large number of assembled viral sequences that have already been produced and will be produced in future studies, making comprehensive manual curation unfeasible. Such quality standards should be able to identify chimeric sequences produced by mis-assembly, incomplete sequences (see also below), and other sources of error, such as PCR/sequencing artifacts, each constituting a large challenge for both conventional metagenomics-based virus discovery and DDVD. Ideally, quality control of viral genome sequences and fragments assembled in DDVD studies will advance current standards [34] and may involve new approaches and metrics to make it more accessible to researchers that are not experts in the field. In the latter respect, we have proposed [24] a system similar to that used by the Protein Data Bank (PDB), where submitters and users can assess an entry via its percentile rank relative to other entries present in the database and with respect to different metrics [35]. For DDVD assemblies, an initial reference set could be seeded with published viral genome assemblies from conventional virus discovery studies, which already went through rigorous, often manual quality control, and gradually extended by DDVD sequences. We have proposed two metrics assessing the per-base and contig-wide accuracy of viral genome assemblies [24], but the actual list of employed metrics could be subject to future updates, and we envision that research groups active in DDVD will develop additional metrics. A quality-control system as described above will enable the identification of outliers of questionable quality via their position in the lowest percentile rank(s), and these outliers might then be analyzed manually in more detail if needed.

### 3.2. The Value of Incomplete Viral Sequences

Another challenge that DDVD has in common with many conventional metagenomics-based virus discovery studies is brought by the considerable fraction of viral genome sequences that are not coding-complete [23,24,25]. For simplicity, we disregard untranslated viral genome regions here when talking about completeness, while being aware of the fact that such regions may encode for important cis-acting regulatory motifs. The challenge of genome incompleteness concerns individual viral nucleic acid sequences, as well as the number of genome segments in the case of segmented and multipartite viruses enclosing their segments in the same and separate capsids, respectively. One typical reason for fragmented viral genome assemblies is poor or absent local read coverage breaking the assembly process, which, in turn, is due to the fact that most of the original studies submitting their data to the SRA or similar repositories were not concerned with virus research and thus did not enrich for viral sequences prior to sequencing. As incomplete genomes do not allow for the determination of the full proteome content of a virus, it may be argued that they are of limited value. They might, for instance, not qualify for taxonomic classification [36,37]. We note, however, that ignoring incomplete viral genome sequences would mean discarding valuable information that could potentially tag unknown viral diversity. It is worth remembering expressed sequence tags (ESTs) and the impact they had on gene discovery before the human genome sequence was available [38]. We anticipate a comparable impact of incomplete viral genome sequences on virus discovery and diversity research in such a way that they may constitute important stepping stones on the way toward a possibly comprehensive description of the global virome.

### 3.3. Advancing Assembly Approaches and Tools

The relatively large number of incomplete genome sequences produced by DDVD studies highlights the need to develop advanced and tailored sequence assembly tools. Specifically, such assemblers would be able to cope with strongly varying and locally poor read coverage, as this constitutes a major reason for fragmented assembly results. Another challenge associated with viral genome assembly, particularly in the case of RNA viruses, is an extraordinarily high sequence variability that is regularly observed at hypervariable regions of viral genomes, which usually cannot be resolved by standard sequence assemblers. In our experience, manual intervention during the assembly process can help to overcome such coverage gaps and variability hotspots of viral quasispecies divergence, suggesting that it may be feasible to design new algorithms for improved automated viral genome assembly. For instance, assembling based on the encoded protein sequence [39] would be an approach to counteract high variability at the nucleotide level, which otherwise leads to fragmentary contig sequences. Moreover, one could envision a strategy that utilizes complete genome sequences of (relatively close) relatives of the viral genome to be assembled in order to conduct a scaffolding of genome fragments by estimating their offset via the offset of homologous parts in the related genome(s). Depending on the degree of relatedness and the strength of sequence conservation, it might even be possible to infer the identity of missing sequences or sequence motifs. Such a hybrid approach, combining canonical reference-based and de novo assembly strategies, may help to further expand our knowledge of viral genetic diversity. Another promising direction that could be followed is to combine data sets positive for strains of the same virus, either prior to the sequence assembly process or after the individual assemblies have been produced. We expect such meta-assembly approaches to generate longer contigs and less fragmentary viral genome sequences on average while retaining good assembly quality.

### 3.4. Detection of Highly Divergent Viruses

Besides generating complete viral genome sequences, it will be equally important to further advance the sensitivity of detecting highly divergent viruses. The identification of viral sequences in sequencing data typically relies on the availability of related viral reference genomes and the ability of computational tools to discriminate between similarity due to homology from chance similarity or convergence [18,20,23,24,40], which is especially critical for most distantly related viruses. It is still not fully understood how well available viral reference sequence sets represent the total viral diversity in nature, as recently demonstrated by the discovery of several potentially novel RNA virus phyla of mostly marine origin [22,25], which add to the five RNA virus phyla that had been described before [41]. It will, therefore, be critical for the success of DDVD to continue a thorough description of the virosphere and to unveil prototypes of yet-unknown main viral lineages. Indeed, DDVD’s superior performance is, in part, due to its ability to consider the known virosphere as a reference dataset and founded on massive amounts of data available in sequencing archives. Furthermore, a promising approach for the detection of divergent viral sequences, although being computationally expensive, involves the iterative re-analysis of sequencing datasets by utilizing reference sequence sets that are expanded with new viral sequences discovered in prior iterations, which gradually increases the viral diversity captured and thus the sensitivity of the search [22]. Moreover, as demonstrated by homology search approaches utilizing protein secondary structure information [42,43], the continuous advancement of structure prediction methods, such as AlphaFold [44], may allow for developing novel protein tertiary structure-based approaches for remote homology detection, leveraging the fact that structure is typically conserved more strongly than sequence [45]. As it is currently unclear whether existing and proposed methods will be able to scale to the dramatic increases in sequencing data expected in the coming years, it might be necessary to also develop fundamentally novel approaches for the detection of highly divergent viral sequences. Such methods could be homology-independent and may examine various sequence features. One such feature could be the proportion of a sequence that is protein-coding, which is usually very high, especially for RNA virus and certain DNA virus genomes, while being comparably low for genomic sequences of cellular organisms.

## 4. Paradigms for Publication of Data-Driven Virus Discovery Results

### 4.1. Upgrading the Product of Data-Driven Virus Discovery

The advent of large-scale DDVD studies poses the question of whether the community is in need of new paradigms for publishing viral sequences discovered in silico by analyzing sequencing archives. We encourage the community to discuss the value and significance of the product generated, i.e., the assembled viral sequences, in light of the fact that the raw data have been generated by others. It is without question that the authors of the original studies in which the data were produced must be properly cited and acknowledged, which also includes linking between respective database entries. Likewise, it is, in our opinion, equally important to recognize that DDVD researchers generate new knowledge from these data that would, in all likelihood, have remained hidden otherwise. Many of the newly assembled viral sequences are not present in any public database, except in the form of raw reads in the respective sequencing repository. We, therefore, argue that it would be counterproductive to designate DDVD results as being secondary data. Currently, DDVD researchers are confronted with certain hurdles when publishing their results, for instance, during submission to NCBI/GenBank. It is typically required to submit the viral genome sequences to a special database division called Third Party Annotation (TPA), seemingly downgrading their status to some extent. As the massive inflow of viral sequences from DDVD studies will by no means stop in the foreseeable future, we encourage the community to discuss whether this approach is still timely.

### 4.2. Evidence of Virus Presence and Identity

An even more pressing question, in our opinion, relates to the type of evidence that should be required for accepting the discovery of a bona fide virus. This concerns viral replication competence and particle formation, but also the packaging of different genome segments in the case of segmented and multipartite viruses. Classical virology would request to demonstrate each of these activities by respective wet-lab experiments. DDVD researchers, however, typically do not have access to the original specimens, and data submitters usually do not store their samples for longer periods of time, making canonical “biological validation” an unreachable goal in the vast majority of cases. We, therefore, encourage the virological community to (re)consider comparative genomics and phylogenetics as providing sufficient evidence for viral presence and identity. Assuming adequate sequence assembly quality (see above), the detection of homologous viral protein domains, segregated in a specific order, in an assembled sequence via sequence comparison with often well-characterized reference viruses makes it highly unlikely that the discovered sequence is anything else than of viral origin. Properties, for instance the enzymatic activity of viral polymerases, including conserved catalytic residues, can thus be accurately inferred for the newly discovered virus (genome) by transferring functional annotations from experimentally characterized reference viruses. Adding secondary or tertiary protein structure information, which can now be computationally predicted with high accuracy [44], to the analysis can provide further evidence, for instance, for annotating divergent viral structural proteins. Additionally, phylogenetic analysis can be used to study gene gain and loss along evolutionary trajectories. Indeed, there are sufficient examples, where subsequent wet-lab research has proven the functionality of viruses discovered by in silico data mining, e.g., regarding their replication competence, genome replication mechanism, ultrastructural features, and virus–host interactions [18,46,47,48]. This demonstrates how classical (molecular) virology can benefit from the product generated by DDVD.

Another important problem is posed by viral sequences integrated into eukaryotic host genomes, often termed endogenous viral elements (EVEs). These can often be discriminated from genomes of extant viruses by flanking host sequences and the accumulation of nonsense and frameshift mutations [18,49,50,51,52], unless only a few reads without flanking host sequences are present in the data, in which case their origin (endogenous or exogenous) may remain undetermined. A similar challenge is presented by temperate bacteriophages that integrate their genome into the host genome during an obligatory prophage stage in their life cycle, but one could argue that sequences from either the latent (prophage) or virulent form are sufficient to determine virus identity, and bioinformatics tools for prophage discovery are available [53].

Comparative genomics and phylogenetics, therefore, provide scalable and accurate tools that can cope with the constantly increasing number of sequences from DDVD studies and can be used to validate and characterize this invaluable source of new knowledge.

## 5. Conclusions and Future Perspectives

We observe that public repositories of (unprocessed) Next-Generation Sequencing (NGS) data are growing at a more–than-linear rate, with the vast majority of NGS projects being unrelated to virological research questions. If the specimen subjected to NGS was infected by a virus at the time of sampling, viral genomes will be sequenced as unrecognized by-catch.

The abovementioned developments prompted the emergence of Data-Driven Virus Discovery (DDVD), which is not primarily technology-driven, but rather data-centered and computational, is uncoupled from the collection and processing of biological samples, exploits publicly and freely accessible data from sequencing archives in high-throughput, and utilizes cluster- or cloud-based high-performance computing. Furthermore, DDVD requires the development of community-wide quality standards, highlights the need for tailored methods and approaches for viral sequence assembly, homology detection, and host assignment, and emphasizes the value of partial viral genome sequences. DDVD also challenges paradigms for required evidence of virus presence and identity, and demands for discussing adjusted (sequence) publication policies.

DDVD brings benefits for virology and molecular biology as it offers a promising avenue toward a possibly comprehensive description of the virosphere, can connect in silico and wet-lab functional research in virology, offers a way to address the uncontrolled presence of viruses in a sample to account for unexplained variance in the outcome of experiments in molecular biology, and can be generalized to all biological entities in a sample, including bacteria, other microorganisms, and other selfish genetic elements.

## Figures and Tables

**Figure 1 biomolecules-12-01073-f001:**
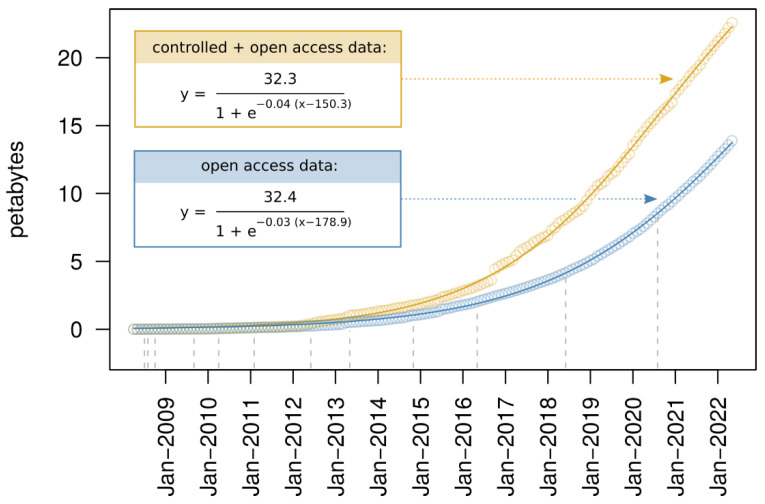
Size increase in the Sequence Read Archive. Shown is the cumulative amount of the total (yellow) and open access (blue) petabytes deposited in the SRA for each month between April 2008 and May 2022. The points represent the actual amounts and the solid lines show the nonlinear least squares fits of logistic functions that captured the trend of the nonlinear increase considerably better than exponential functions (not shown). Parameters of the fitted curves are detailed in the inlets. The dashed vertical lines indicate the time points at which the amount of open access data doubled relative to the previous doubling time point.

## Data Availability

The data used to produce Figure 1 is freely available on the NCBI’s SRA website (www.ncbi.nlm.nih.gov/sra/docs/sragrowth/ accessed on 29 June 2022).

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
