# Peer review of "Opportunities and Challenges of Data-Driven Virus Discovery"

_biomolecules, 2022, doi:10.3390/biom12081073_

Round 1

Reviewer 1 Report

The paper gives an overview of early laboratory-based work in virus discovery before moving onto describe computational-based Data-Driven Virus Discovery (DDVD).

Overall the paper is quite well balanced. I am not sure who the intended
audience is, though. Most people working in virology, including
non-computational people, will be aware of many of the discussed issues
(but that is just my opinion and it could well be incorrect).

My main comment is that I think the paper would be more valuable if it gave more information about the challenges that the computational approach faces right now and going forward and less feel-good optimism which isn't really helpful, it just paints a nice picture of a possible sunny long-term future.

For example, some challenges that could be described:

  - The issue of identifying highly divergent viruses isn't going to be
    solved by just making incrementally larger databases. The authors
    mention approaches using protein folding, but can these be expected to
    scale to examining millions or billions of sequencing reads? We need
    some fundamental changes here, and it's not clear what they will be.

  - The issue of contamination might be considered. In our setting we
    identified 18 different steps where contamination (not to mention
    sources of confusion or error) could be introduced before FASTQ
    sequencing files are available for processing. This introduces problems
    that can be very difficult to detect and resolve even when you do have
    the samples around and the kits and all information about what took
    place.

  - What about other sources of error? PCR, sequencing, etc.

  - Mis-assembly is a big challenge for metagenomics. That could just mean chimeric sequences, but also, as the authors note, issues of incomplete segments or the challenge of deciding what segments correspond to what virus in a metagenomic data set.

  - The issue of integrated viruses is glossed over by saying that flanking
    DNA will save the day, but what about finding small numbers of reads
    with no flanking DNA? This leads to great difficulty.

Aspects that are overly Panglossian:

  - "Taking this reasoning another step forward, DDVD could stimulate the
    field to develop standards that would empower sequence coverage and
    depth for all organisms in a biological sample, including bacteria and
    other microorganisms".  I find this very naively optimistic. How do you
    "empower sequence coverage and depth" anyway? The sequencing of     even simple other eukaryotic or bacterial organisms can be vastly more
    complicated than that of even the most complicated virus.

  - I would delete "Analysis of the data will also not induce additional
    costs if scientific high-performance computers are utilized or assuming
    that charges normally made by cloud computing providers can be
    circumvented". There is no free lunch.

  - "they may constitute important anchor points on the way towards a
     comprehensive description of the global virome." Maybe "stepping
     stones" would be better. And do you really expect there will ever be a
     comprehensive description of the global virome? In what year?

Minor comments
--------------

The sentence "The accumulation of new sequencing experiments in the SRA during recent years has proceeded at a rate that exceeds linear growth." is strange.  Exceeding linear growth is an extremely mild way to describe the growth. Can this be more quantitative, with a few numbers?

Section 2.3. Add "economically important" as a regularly investigated
category of virus. TMV wasn't the first virus discovered just by chance,
and much of what we know about animal viruses comes from studying things that have an economic impact on domesticated or farmed animals.

Section 3.3 mentions "variability hotspots" - how is this to be differentiated from quasispecies and minority variants?

Trivial comments
----------------

Font change in the first paragraph of section 2.1

"sequences from viruses can be detected in nucleotide archives as a
by-product of sequencing the host". Could be more nuanced, and mention the possibility of integrated viruses.

The suggestion that viral metagenomics could show the way for sequencing of all organisms is a bit optimistic or even naive.

Could say more about how many things that we now know about microbiology have come from the study of viruses.

Section 2.4 "to unveil cases" -> "to detect cases"

End of section 2.4, I would delete "This list is subject to further
refinement by the community in the future." It's a bit tautological /
obvious, and not for them to say.

Section 2.5: "that shall not be overlooked" -> should not.

Section 2.5: "re-used freely and accessed publicly" what is the difference?
Do you really need to say both?

This is a strange phrasing: "outcompeting many other types of data".

"but the actual list of employed metrics could be subject to future
updates, and we envision that research groups active in DDVD will develop
additional metrics" Is a bit verbose, and also obvious.

"In this respect we would like to remind about" -> It is worth remembering...

Font change at end of section 3.3

Section 3.4 "which may multiplicate the five RNA virus phyla" - very awkward.

Section 4.1. Valorizing the product of Data-Driven Virus Discovery
Is valorizing a word?

Section 4.2: And phylogenetic analysis can be used to rationalize gene gain and loss along evolutionary trajectories. What does "rationalize" mean here?

Author Response

We thank the four reviewers for providing helpful feedback to our manuscript. Below we give a point-by-point response to the issues raised. The points brought up by the reviewers and our response are in italic and regular font, respectively.

Reviewer #1:

R1.1. The paper gives an overview of early laboratory-based work in virus discovery before moving onto describe computational-based Data-Driven Virus Discovery (DDVD). Overall the paper is quite well balanced. I am not sure who the intended audience is, though. Most people working in virology, including non-computational people, will be aware of many of the discussed issues (but that is just my opinion and it could well be incorrect).

We thank the reviewer for the overall positive evaluation of our manuscript.

R1.2. My main comment is that I think the paper would be more valuable if it gave more information about the challenges that the computational approach faces right now and going forward and less feel-good optimism which isn't really helpful, it just paints a nice picture of a possible sunny long-term future.

For example, some challenges that could be described:

- The issue of identifying highly divergent viruses isn't going to be solved by just making incrementally larger databases. The authors mention approaches using protein folding, but can these be expected to scale to examining millions or billions of sequencing reads? We need some fundamental changes here, and it's not clear what they will be.

- The issue of contamination might be considered. In our setting we identified 18 different steps where contamination (not to mention sources of confusion or error) could be introduced before FASTQ sequencing files are available for processing. This introduces problems that can be very difficult to detect and resolve even when you do have the samples around and the kits and all information about what took place.

- What about other sources of error? PCR, sequencing, etc.

- Mis-assembly is a big challenge for metagenomics. That could just mean chimeric sequences, but also, as the authors note, issues of incomplete segments or the challenge of deciding what segments correspond to what virus in a metagenomic data set.

- The issue of integrated viruses is glossed over by saying that flanking DNA will save the day, but what about finding small numbers of reads with no flanking DNA? This leads to great difficulty.

We thank the reviewer to pointing to these important aspects. It is not our intention to “paint a nice picture of a possibly sunny long-term future”, which is why we devoted approximately one-third of the manuscript to challenges associated with DDVD. We have now further expanded this section to address the five specific points raised by the reviewer.

Aspects that are overly Panglossian:

R1.3. "Taking this reasoning another step forward, DDVD could stimulate the field to develop standards that would empower sequence coverage and depth for all organisms in a biological sample, including bacteria and other microorganisms". I find this very naively optimistic. How do you "empower sequence coverage and depth" anyway? The sequencing of even simple other eukaryotic or bacterial organisms can be vastly more complicated than that of even the most complicated virus.

We fully agree that obtaining full-length genome sequences of eukaryotic and bacterial organisms is an extremely challenging task. But, as explained in the manuscript, it may be sufficient for certain questions to retrieve and taxonomically classify sequence fragments. These incomplete sequences could be considered as tags for the organisms present in a sample. We have adjusted the sentence in question, which is intended to give a possible future perspective, and hopefully simplified its message and removed any naive optimisms.

R1.4. I would delete "Analysis of the data will also not induce additional costs if scientific high-performance computers are utilized or assuming that charges normally made by cloud computing providers can be circumvented". There is no free lunch.

We prefer to keep this sentence as is. As demonstrated by the Serratus team (see their github web page), cloud computing providers like Amazon may offer significant discounts to scientific projects or even do not charge for their web services.

R1.5. "they may constitute important anchor points on the way towards a comprehensive description of the global virome." Maybe "stepping stones" would be better. And do you really expect there will ever be a comprehensive description of the global virome? In what year?

The reviewer asks an interesting question that we cannot answer. However, in our opinion, this does not mean that the scientific community should not aim for a comprehensive description of the virosphere. We have adjusted this sentence as suggested and toned it down by adding the word “possibly”.

Minor comments

--------------

R1.6. The sentence "The accumulation of new sequencing experiments in the SRA during recent years has proceeded at a rate that exceeds linear growth." is strange. Exceeding linear growth is an extremely mild way to describe the growth. Can this be more quantitative, with a few numbers?

We have adjusted this sentence, now mentioning some numbers, and we have also introduced Fig. 1 to visualize the SRA size increase.

R1.7. Section 2.3. Add "economically important" as a regularly investigated category of virus. TMV wasn't the first virus discovered just by chance, and much of what we know about animal viruses comes from studying things that have an economic impact on domesticated or farmed animals.

We have modified this sentence as suggested.

R1.8. Section 3.3 mentions "variability hotspots" - how is this to be differentiated from quasispecies and minority variants?

We now explicitly mention viral quasispecies in this sentence.

R1.9. Trivial comments

----------------

Font change in the first paragraph of section 2.1

"sequences from viruses can be detected in nucleotide archives as a by-product of sequencing the host". Could be more nuanced, and mention the possibility of integrated viruses.

The suggestion that viral metagenomics could show the way for sequencing of all organisms is a bit optimistic or even naive.

Could say more about how many things that we now know about microbiology have come from the study of viruses.

Section 2.4 "to unveil cases" -> "to detect cases"

End of section 2.4, I would delete "This list is subject to further refinement by the community in the future." It's a bit tautological / obvious, and not for them to say.

Section 2.5: "that shall not be overlooked" -> should not.

Section 2.5: "re-used freely and accessed publicly" what is the difference? Do you really need to say both?

This is a strange phrasing: "outcompeting many other types of data".

"but the actual list of employed metrics could be subject to future updates, and we envision that research groups active in DDVD will develop additional metrics" Is a bit verbose, and also obvious.

"In this respect we would like to remind about" -> It is worth remembering...

Font change at end of section 3.3

Section 3.4 "which may multiplicate the five RNA virus phyla" - very awkward.

Section 4.1. Valorizing the product of Data-Driven Virus Discovery. Is valorizing a word?

Section 4.2: And phylogenetic analysis can be used to rationalize gene gain and loss along evolutionary trajectories. What does "rationalize" mean here?

We thank the reviewer for noticing all these minor issues and for providing suggestions for improvement. We have addressed almost all of these points in the revised manuscript.

Reviewer 2 Report

In this manuscript Lauber and Seitz review the current status of computational and data-centered approaches for virus discovery. These sequence data-centered and computational approaches have the potential to impact a number of research areas in virology in unprecedented detail and depth.

The review is presented in a logical manner and provided adequate detail. I gained a good basic understanding of the current status of the field, both in terms of opportunities and challenges and shortcomings.

Overall, this was an interesting review and a well-organized manuscript. I have no serious concerns or any major issues with the manuscript in general.

However, a few minor concerns need to be addressed:

For the conclusions and Future directions section, it may be easier for the reader if the three bulleted sections are converted into three paragraphs with short sentences. In the present form some sentences feel a little clunky or incomplete.

The formatting of the review is inconsistent in terms of font size as well as style. Please ensure that the text is in accordance to the formatting guidelines of the journal.

Author Response

We thank the four reviewers for providing helpful feedback to our manuscript. Below we give a point-by-point response to the issues raised. The points brought up by the reviewers and our response are in italic and regular font, respectively.

Reviewer #2:

R2.1. In this manuscript Lauber and Seitz review the current status of computational and data-centered approaches for virus discovery. These sequence data-centered and computational approaches have the potential to impact a number of research areas in virology in unprecedented detail and depth.

The review is presented in a logical manner and provided adequate detail. I gained a good basic understanding of the current status of the field, both in terms of opportunities and challenges and shortcomings.

Overall, this was an interesting review and a well-organized manuscript. I have no serious concerns or any major issues with the manuscript in general.

We thank the reviewer for the positive evaluation of our manuscript.

However, a few minor concerns need to be addressed:

R2.2. For the conclusions and Future directions section, it may be easier for the reader if the three bulleted sections are converted into three paragraphs with short sentences. In the present form some sentences feel a little clunky or incomplete.

We have converted the bullet points into sentences as suggested.

R2.3. The formatting of the review is inconsistent in terms of font size as well as style. Please ensure that the text is in accordance to the formatting guidelines of the journal.

The formatting was done, presumably automatically, by the journal. We have now hopefully corrected all formatting issues in the revised manuscript.

Reviewer 3 Report

The authors present an interesting perspective article that explores the potential of existing masses of next-generation sequencing data in global repositories to be used for the discovery of novel viruses. They explore the various ways these data might be used and what developments are likely to be required to maximise the gains in this area. Overall the manuscript is well written and the perspectives of the author's on the subject matter are clearly articulated. The subject matter should be of interest to those in several areas from conventional wet-lab virology to those developing new computational strategies to maximise the acquisition of meaning outputs from the large-scale sequencing efforts that are underway globally.

Author Response

We thank the four reviewers for providing helpful feedback to our manuscript. Below we give a point-by-point response to the issues raised. The points brought up by the reviewers and our response are in italic and regular font, respectively.

Reviewer #3:

R3.1. The authors present an interesting perspective article that explores the potential of existing masses of next-generation sequencing data in global repositories to be used for the discovery of novel viruses. They explore the various ways these data might be used and what developments are likely to be required to maximise the gains in this area. Overall the manuscript is well written and the perspectives of the author's on the subject matter are clearly articulated. The subject matter should be of interest to those in several areas from conventional wet-lab virology to those developing new computational strategies to maximise the acquisition of meaning outputs from the large-scale sequencing efforts that are underway globally.

We thank the reviewer for the positive and encouraging evaluation of our manuscript.

Reviewer 4 Report

I think authors made a good job analyzing many advantages and disadvantages of proposed Data-Driven Virus Discovery approach. And I see no reason to repeat some of them.

But, in my understanding this is not "Virus Discovery" by itself but "Discovery of novel viral sequences". And from this point the definition of both terms: Virus and Viral sequence in Introduction section will be useful to understand what authors really mean then using this terms. Also for me is not clear how authors plan to avoid appearing of "fake viruses" if it discovery will based only on Data, because this data can by artificially generated without sequencing.

Author Response

We thank the four reviewers for providing helpful feedback to our manuscript. Below we give a point-by-point response to the issues raised. The points brought up by the reviewers and our response are in italic and regular font, respectively.

Reviewer #4:

R4.1. I think authors made a good job analyzing many advantages and disadvantages of proposed Data-Driven Virus Discovery approach. And I see no reason to repeat some of them.

We thank the reviewer for the overall positive evaluation of our manuscript.

R4.2. But, in my understanding this is not "Virus Discovery" by itself but "Discovery of novel viral sequences". And from this point the definition of both terms: Virus and Viral sequence in Introduction section will be useful to understand what authors really mean then using this terms.

In our opinion, this discrimination of virus and viral sequences and the current standard practice of discovering novel viruses through their genome sequences is already sufficiently well explained in the manuscript.

R4.3. Also for me is not clear how authors plan to avoid appearing of "fake viruses" if it discovery will based only on Data, because this data can by artificially generated without sequencing.

We agree that artificial viral sequences could in principal be generated rather easily, but we do not consider this to be of relevance for our article as the SRA has the requirement to link a submission to a BioSample and because we trust that scientists would not intentionally submit “fake data” to this and similar databases. We therefore prefer not to discuss this aspect in the manuscript.